# Role for formin-like 1-dependent acto-myosin assembly in lipid droplet dynamics and lipid storage

Simon G. Pfisterer[1,2], Gergana Gateva[3], Peter Horvath[4,5], Juho Pirhonen[1,2], Veijo T. Salo[1,2], Leena Karhinen[1], Markku Varjosalo[3], Samppa J. Ryhänen[6], Pekka Lappalainen[3] & Elina Ikonen[1,2]

Lipid droplets (LDs) are cellular organelles specialized in triacylglycerol (TG) storage undergoing homotypic clustering and fusion. In non-adipocytic cells with numerous LDs this is balanced by poorly understood droplet dissociation mechanisms. We identify non-muscle myosin IIa (NMIIa/MYH-9) and formin-like 1 (FMNL1) in the LD proteome. NMIIa and actin filaments concentrate around LDs, and form transient foci between dissociating LDs. NMIIa depletion results in decreased LD dissociations, enlarged LDs, decreased hydrolysis and increased storage of TGs. FMNL1 is required for actin assembly on LDs *in vitro* and for NMIIa recruitment to LDs in cells. We propose a novel acto-myosin structure regulating lipid storage: FMNL1-dependent assembly of myosin II-functionalized actin filaments on LDs facilitates their dissociation, thereby affecting LD surface-to-volume ratio and enzyme accessibility to TGs. In neutrophilic leucocytes from MYH9-related disease patients NMIIa inclusions are accompanied by increased lipid storage in droplets, suggesting that NMIIa dysfunction may contribute to lipid imbalance in man.

[1] Department of Anatomy and Research Programs Unit, Faculty of Medicine, University of Helsinki, Helsinki 00290, Finland. [2] Minerva Foundation Institute for Medical Research, Helsinki 00290, Finland. [3] Institute of Biotechnology, University of Helsinki, Helsinki 00790, Finland. [4] Institute for Molecular Medicine Finland (FIMM), University of Helsinki, Helsinki 00290, Finland. [5] Synthetic and Systems Biology Unit, Hungarian Academy of Sciences, BRC, Szeged H-6726, Hungary. [6] Division of Hematology-Oncology and Stem Cell Transplantation, Children's Hospital, Helsinki University Central Hospital and University of Helsinki, Helsinki 00290, Finland. Correspondence and requests for materials should be addressed to E.I. (email: Elina.Ikonen@helsinki.fi).

In cells, lipid droplets (LDs) are highly dynamic, well-connected with other organelles and actively communicating with one another[1–3]. LDs undergo major morphological changes depending on the metabolic status of cells. In energy surplus, stimulation of triacylglycerol (TG) storage results in LD enlargement, apparently by both fusion and lipid transfer between adjacent LDs[4,5], with the extreme phenotype being unilocular LDs in mature adipocytes. The homotypic fusion of LDs is regulated by Fsp27 enriched at LD contact sites and its activity is controlled by Rab8a[6]. Conversely, TG breakdown by lipolysis is accompanied by LD shrinkage, dissociation and dispersion[7]. This process involves modification of LD resident proteins to allow docking and activation of lipolytic enzymes to digest the lipid core[8], and eventual dispersion of smaller LDs to the cell periphery in a microtubule-dependent fashion[7,9]. How the initial dissociation of adjacent LDs is achieved, is essentially unknown.

Actin and actin-binding proteins have been reported to be associated with LDs[10–12]. In murine macrophages, pharmacological disruption of the actin cytoskeleton reduces LD size and lipid storage[10] and in 3T3-L1 cells, inhibition of actin dynamics through cofilin-1 depletion disrupts adipogenesis and lipid storage[13]. At the organismal level, pharmacological perturbation of acto-myosin turnover in zebrafish affects the recruitment of LDs from the yolk-blastodisc to the animal pole[14]. However, whether LDs associate with specific actin filament structures and what functional role they may have, has not been defined.

Here, we provide evidence for the presence of a novel acto-myosin structure that associates with LDs to control their dynamics. We show that the actin-binding protein formin-like 1 (FMNL1) facilitates the assembly of non-muscle myosin IIa (NMIIa) containing actin filament arrays on LDs. Importantly, depletion of NMIIa from human cells reduces LD dynamics and alters their clustering propensity, resulting in supersized LDs and reduced TG hydrolysis. Together, these data suggest that NMIIa assembling with actin in focal points around LDs assists in the dissociation of clustered LDs.

## Results

**NMIIa depletion results in enlarged LDs and increased TGs**. We conducted a proteomic profiling of LD-associated proteins. This revealed several actin-binding proteins (Supplementary Table 1), including NMIIa (encoded by the MYH9 gene), a structural component of contractile actin filaments. NMIIa has earlier been identified in the LD proteome of several organisms[10–12], but its functional role at LDs remains unknown.

To investigate the role of NMIIa at LDs, we silenced the protein from human osteosarcoma U2OS cells (Supplementary Fig. 1a–d), a cell line commonly used for studying cytoskeletal dynamics. For systematic analysis of LDs in U2OS cells, we developed automated image analysis tools to quantify LD abundance, size and clustering (Supplementary Fig. 2a). Under normal growth conditions, roughly 100 LDs were detected in these cells using fluorescent LD dyes, with small LDs (below $0.5\,\mu m^2$) representing the overwhelming majority (Fig. 1a,b, Supplementary Fig 2b,c). Oleic acid administration for 7 h roughly doubled the number of LDs (Supplementary Fig. 2b), whereas their relative size distribution was not changed (Supplementary Fig. 2c). Interestingly, depletion of NMIIa resulted in a LD rearrangement with larger LDs being significantly increased (about sixfold for LDs between 0.5 and $1.0\,\mu m^2$ and ninefold for LDs in the range of $1.5–2\,\mu m^2$; Fig. 1a,b). A similar increase in LD size was observed upon oleic acid administration in NMIIa-depleted cells (Fig. 1c,d). Enhanced formation of large LDs was also observed with another NMIIa-targeting siRNA, suggesting that the effect is specific (Supplementary Fig. 2d).

To further assess the specificity of the LD enlargement upon NMIIa depletion, we generated U2OS cells stably overexpressing an siRNA-resistant Cherry-NMIIa, or Cherry alone as control (Fig. 1e,f). We found that the accumulation of enlarged LDs upon NMIIa depletion was rescued by Cherry-NMIIa overexpression in control conditions (Fig. 1e,f) as well as upon oleic acid loading (Supplementary Fig. 2e). Conversely, NMIIa overexpression in control cells resulted in the reduction of large LDs (Fig. 1f). In addition, acute inhibition of NMIIa ATPase activity for 1 h with blebbistatin[15] showed a similar tendency toward enlarged LDs (Supplementary Fig. 2f). Together, these data provide additional evidence that the LD enlargement is a specific effect of NMIIa loss.

We next explored whether depletion of NMIIa affects neutral lipid storage. We found that the larger size of LDs in NMIIa-depleted cells was accompanied by a significant increase in TG deposition compared to control cells (Fig. 1g). A similar effect was obtained upon NMIIa inhibition with blebbistatin (Fig. 1h). The increased TG content was due to impaired TG breakdown rather than increased TG synthesis, as in NMIIa-silenced cells, TG hydrolysis was reduced (Fig. 1i) but TG synthesis was unaffected (Fig. 1j). This is in line with the idea that the lower surface-to-volume ratio in larger LDs reduces the accessibility of hydrolytic enzymes to droplets[16].

Importantly, general disruption of NMIIa-positive stress fibres through inhibition of the small GTPase Rho in filopodia or silencing of tropomyosin 4 (ref. 17) did not increase the proportion of large LDs in either control or oleic acid loaded conditions (Supplementary Fig. 3). These data argue that the observed LD phenotypes in NMIIa-depleted cells are not due to problems in stress fibre assembly, but rather that NMIIa contributes to LD dynamics through a more specific mechanism.

**NMIIa localizes transiently to LD dissociation sites**. We investigated the localisation of NMIIa at LDs using confocal and three-dimensional structured illumination microscopy (3D-SIM). To facilitate this, we employed mild saponin permeabilization of cells prior to fixation, to reduce the cytoplasmic pool of NMIIa (Supplementary Fig. 1e,f). We found that endogenous NMIIa localized to focal points around LDs visualized by the LD marker peptide HPOS[18]-Cherry stably expressed in U2OS cells (Fig. 2a,b), whereas the periodic labelling pattern of NMIIa along stress fibres was preserved (Fig. 2b). The LD-associated NMIIa patches often located between two neighbouring LDs or between clustered LDs (Fig. 2a,b, Supplementary Fig. 1g–k). On average, LDs were associated with three NMIIa patches and approximately one patch per LD–LD contact site (Supplementary Fig. 1g,h). Also filamentous actin (F-actin) was found at LDs and LD–LD contact sites and a subset of this actin colocalized with endogenous NMIIa at LDs (Fig. 2b, Supplementary Fig. 1i). For more detailed investigation of NMIIa localisation at LDs we used immuno-electron microscopy. This recapitulated, in addition to the expected periodic NMIIa immunoreactivity along stress fibres, NMIIa on the surface of individual LDs and at LD–LD contact sites (Fig. 2c).

To monitor NMIIa dynamics at LDs in living cells, we used Airyscan superresolution microscopy of GFP-NMIIa-expressing cells. This revealed distinct GFP-NMIIa patches transiently concentrating between clustered LDs (Fig. 2d, Supplementary Movie 1; for additional examples, see Supplementary Fig. 4a, Supplementary Movies 2 and 9). In many cases, transient NMIIa accumulation between two LDs was followed by their dissociation (Fig. 2d, Supplementary Movie 1). Plotting of NMIIa and

LD intensity profiles of dissociating LDs before NMIIa accumulation, upon NMIIa enrichment and after LD segregation, showed a transient increase in NMIIa fluorescence during the partitioning of LDs (Fig. 2e, Supplementary Fig. 4b). Furthermore, we observed transient accumulation of fluorescently tagged actin between dissociating LDs (Supplementary Fig. 4c,d and Supplementary Movie 3), suggesting that NMIIa and actin filaments cooperate in the dissociation of LDs.

**NMIIa affects LD dissociation dynamics and LD clustering.** To address if the loss of NMIIa affects LD dynamics, we monitored the association kinetics of clustered LDs, that is, LDs that were associated with each other. Coherent anti-stokes Raman scattering (CARS) microscopy was employed for label-free imaging of LDs in live cells (Fig. 3a,b, Supplementary Movie 4). This revealed that in control cells, the clustered LDs either stayed associated with each other (static clusters) or were highly

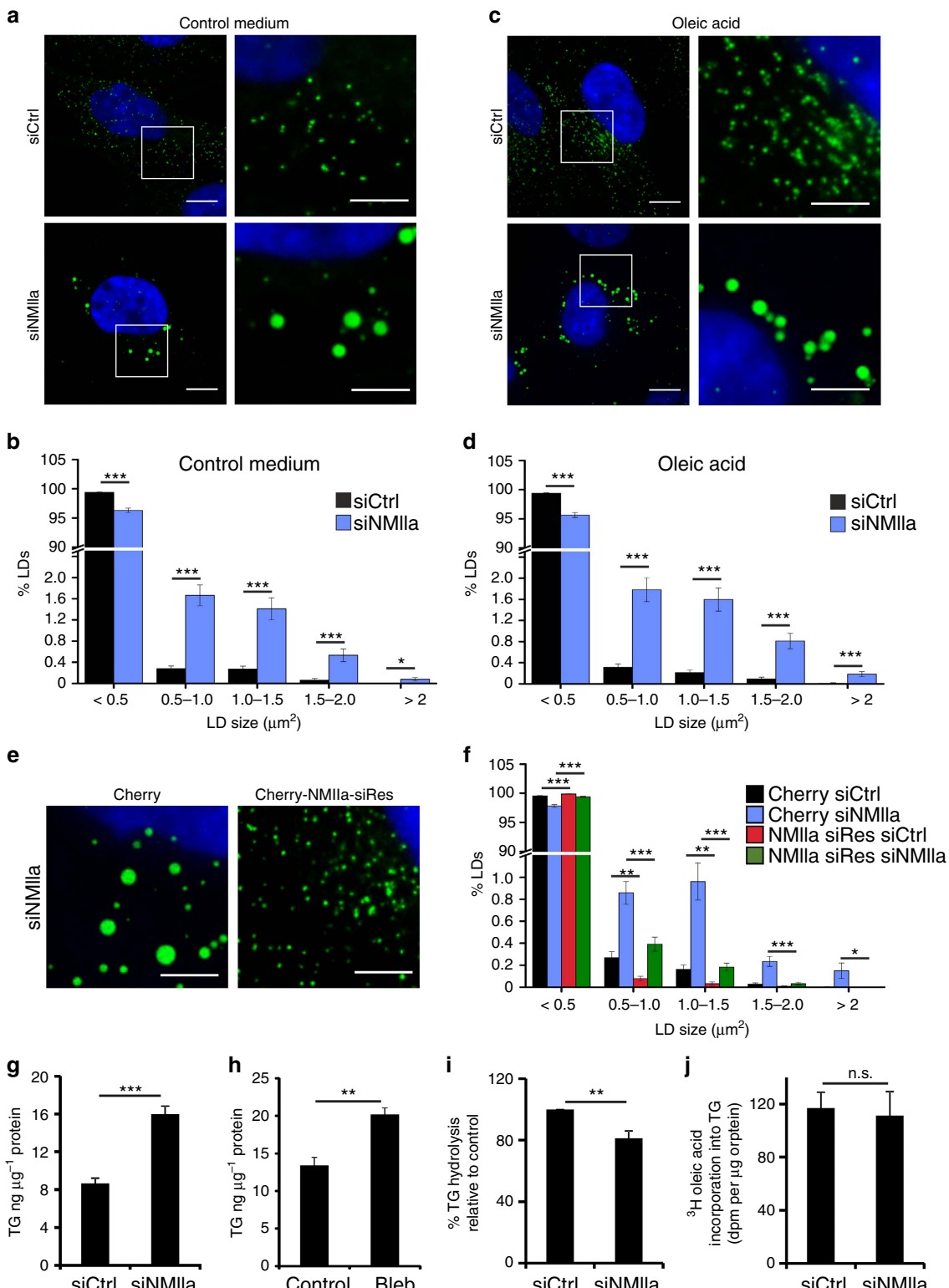

dynamic, with rapid association and dissociation of the clustered LDs during the 5 min recording (Fig. 3a, Supplementary Movie 4). Interestingly, in NMIIa-depleted cells, static LD associations were more prevalent (Fig. 3b, Supplementary Movie 4). Similar observations were made upon NMIIa inhibition with blebbistatin or disruption of actin polymerisation with cytochalasin D (Supplementary Movies 5 and 6). Analogously to blebbistatin, acute cytochalasin D treatment increased the proportion of larger LDs (Supplementary Fig. 2g). Owing to the similar effects of NMIIa depletion and blebbistatin on LDs, the motor activity of myosin II appears functionally important, and other myosin isoforms, including NMIIb, seem not to fully compensate for the loss of NMIIa.

To obtain further quantitative information on the altered LD association characteristics upon NMIIa depletion, we extended the automated LD detection pipeline with a cluster analysis tool (Fig. 3c). This tool considers LDs to be clustered if the distance between droplet boundaries is smaller or equal to two pixels (0.266 μm), enabling the detection of clusters with a variable LD number and quantification of average LD sizes as well as LD size heterogeneity within clusters. In control cells loaded with oleic acid, the majority of LD clusters comprised two droplets, followed by a decreasing abundance of clusters containing more droplets (Fig. 3d). Interestingly, NMIIa-depleted cells loaded with oleic acid contained overall less-clustered LDs, with most pronounced reductions for clusters of three and four LDs (Fig. 3d). In parallel, the mean size of clustered LDs increased by ~75% upon NMIIa silencing (Fig. 3e) and their size heterogeneity increased, as evidenced by greater than threefold increase in the s.d. of the size of clustered LDs (Fig. 3f,g). The observed changes in the size of clustered LDs were due to loss of NMIIa, as they were reversed in cells overexpressing the siRNA-resistant Cherry-NMIIa (Fig. 3h,i). These results indicate that NMIIa depletion results in reduced motility of associated LDs and a larger size heterogeneity between associated LDs. Moderate alterations in the balance of LD dissociation/re-association may over time result in prominent LD size changes, with bigger LDs deriving from coalescence of neighbouring LDs, as suggested by the observed fusion between immotile, clustered LDs (Supplementary Fig. 4e,f, Supplementary Movie 7).

**FMNL1 localizes to LD dissociation sites.** Our proteomic analysis also revealed the actin regulatory protein FMNL1 among the LD-associated proteins (Supplementary Table 1). FMNL1 has several activities, such as stimulation of actin polymerisation and bundling of actin filaments[19,20], providing a candidate to drive the formation of actin filaments at LDs. We therefore investigated the localisation of endogenous FMNL1 in U2OS cells using Airyscan and 3D-SIM imaging. FMNL1 antibodies revealed punctate cytoplasmic as well as perinuclear immunoreactivity, in agreement with previous studies[21]. To reduce cytoplasmic immunoreactivity, cells were saponin permeabilized pre-fixation, similarly as for NMIIa staining (Supplementary Fig. 5d,e). This highlighted punctate FMNL1 labelling on the surface of LDs (Fig. 4a,b). We found on average three FMNL1 patches associated per LD and of these approximately half localized to LD–LD contact sites (Supplementary Fig. 5f,g). Interestingly, FMNL1 patches at LDs were closely apposed to NMIIa dots, as visualized by myosin light chain staining (Fig. 4c, Supplementary Fig. 5h). Live cell Airyscan video microscopy demonstrated that GFP-FMNL1 localized transiently to LD dissociation sites (Fig. 4d, Supplementary Movie 8). Of the 155 LD dissociation events quantified, over 60% showed FMNL1 foci between the dissociating LDs (Fig. 4f) and in nearly all of them, FMNL1 was present already before the separation of LDs occurred (Fig. 4g). When a similar analysis was performed from live cell Airyscan videos of GFP-NMIIa expressing cells, we observed about half of the LD dissociation sites to contain NMIIa accumulations (Fig. 4e,f; Supplementary Movie 9). Interestingly, NMIIa seemed to be recruited to the dissociation sites after FMNL1, as in ~1/3 of the events the NMIIa accumulation appeared once LDs had started to separate from each other (Fig. 4g). We also noted that after dissociation some LDs underwent reassociation and that FMNL1 could stay associated with LDs during such events (Supplementary Fig. 6 Supplementary Movies 10 and 11).

To investigate the interrelationship of FMNL1 and actin upon LD dissociation, we performed triple-colour live cell Airyscan imaging of GFP-FMNL1, BFP-Lifeact and LDs in a region below the nucleus, to minimize cytoplasmic background (Fig. 5a). This revealed a transient focal accumulation of GFP-FMNL1 in a LD cluster, followed by dynamic accrual of BFP-Lifeact between dissociating LDs (Fig. 5b,c; Supplementary Movie 12). We also monitored the accumulation of a larger FMNL1 patch apparently driving the separation of neighbouring LDs (Fig. 5d–f; Supplementary Movie 13). This enabled focal intensity measurements of FMNL1 and Lifeact, further substantiating the idea that an increase in FMNL1 intensity preceeded that of Lifeact between LDs (Fig. 5f).

**FMNL1 drives actin assembly on LDs and recruits NMIIa to LDs.** Together, these data suggest that FMNL1 can generate actin filaments on LDs, which in turn may recruit NMIIa. In support of this idea, we found a major reduction of NMIIa in LD fractions of FMNL1-silenced cells (Fig. 6a–c; Supplementary Fig. 7f,g). This was accompanied by a decrease of NMIIa immunostaining at LDs upon FMNL1 depletion (Supplementary

**Figure 1 | NMIIa depletion results in enlarged LDs and increased TG deposition.** (**a**) LD morphology of U2OS cells treated with control or NMIIa siRNA. Cells were stained with LipidTOX green (LDs), DAPI (nuclei) and CellTracker Red (cytoplasm, not shown). (**b**) Automated LD quantification of U2OS cells treated as in **a**, ± s.e.m., $n = 303$ cells for siCtrl and 228 cells for siNMIIa from three independent experiments. (**c**) Similar as in **a** with 200 μM oleic acid treatment for 7 h. (**d**) Automated LD quantification of cells treated as in **c**. ± s.e.m. $n = 293$ cells for siCtrl and 204 cells for siNMIIa, from three independent experiments. (**e**) LD morphology of U2OS cell lines expressing either Cherry or siRNA resistant Cherry-NMIIa (Cherry-NMIIa siRes) and treated with NMIIa siRNA. Cells were stained with LD540 (LDs), DAPI (nuclei) and CellTracker Red (cytoplasm, not shown). (**f**) Automated LD quantification of cells treated as in **e**, ± s.e.m. $n = 339$ for Cherry siCtrl, 261 cells for Cherry siNMIIa, $n = 303$ cells for Cherry NMIIa siRes siCtrl and 270 cells for Cherry NMIIa siRes siNMIIa, from 3 independent experiments. (**g**) TG amount of U2OS cells treated with siCtrl or siNMIIa ( ± s.e.m., $n = 6$–7 samples from three independent experiments). (**h**) TG amount of U2OS cells treated with or without 30 μM Blebbistatin for 24 h ( ± s.e.m., $n = 5$ samples from two independent experiments). (**i**) TG hydrolysis in U2OS cells treated with siCtrl or siNMIIa. Cells were treated with 200 μM oleic acid overnight to induce TG storage followed by incubation in complete medium for 8 h to induce TG hydrolysis ( ± s.e.m., $n = 8$ samples from four independent experiments). (**j**) TG synthesis as measured by [3]H-oleic acid incorporation for 2 h in serum free conditions in U2OS cells treated with siCtrl or siNMIIa ( ± s.e.m., $n = 6$ samples from two independent experiments). Scale bar 10 μm, 5 μm for insets. Student's t-test, *$P < 0.05$, **$P < 0.01$, ***$P < 0.001$.

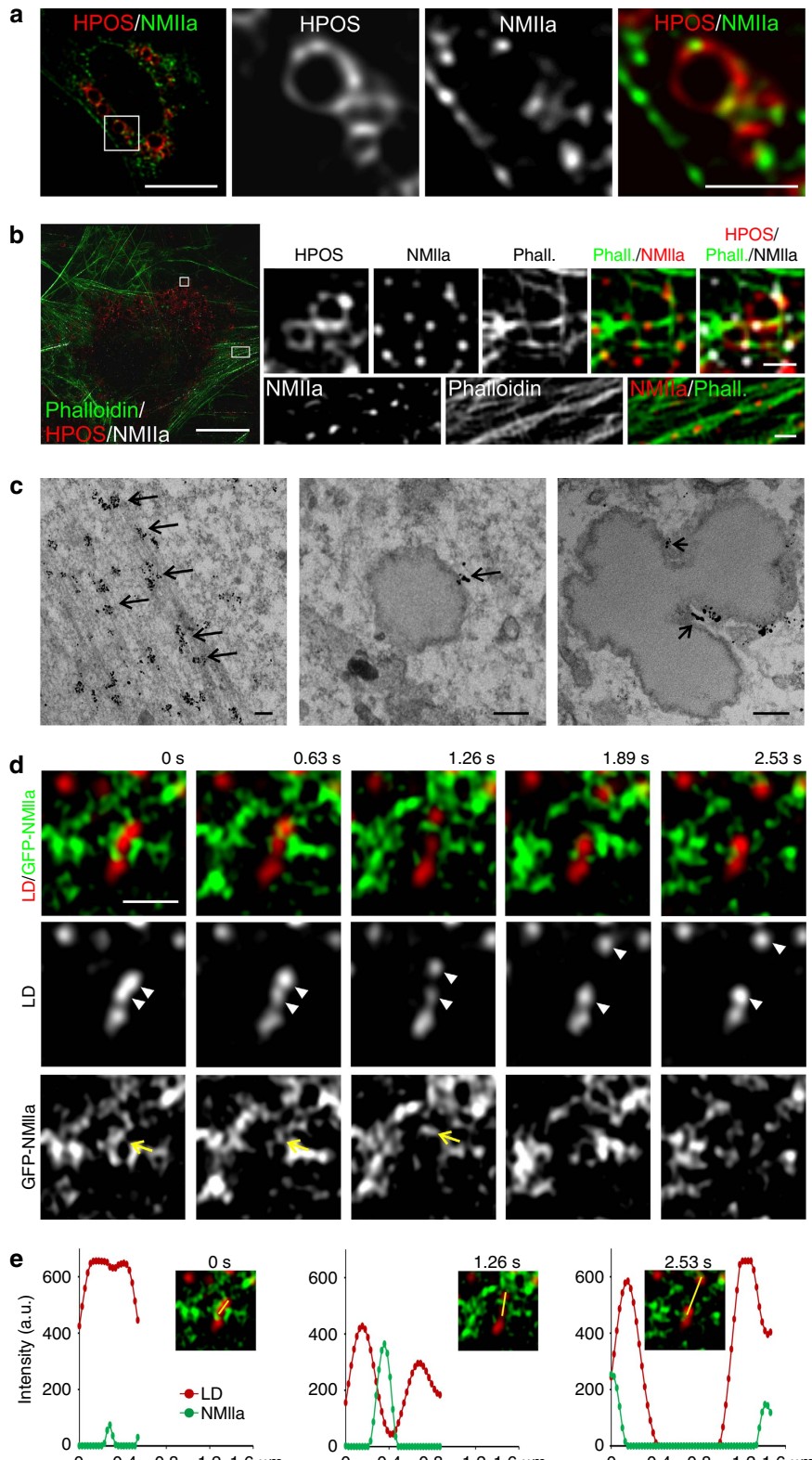

**Figure 2 | NMIIa colocalizes with actin on LDs and concentrates between dissociating LDs.** (**a**) Stable U2OS Cherry-HPOS cells were treated with 400 µM oleic acid overnight, saponin permeabilized, fixed and immunostained for NMIIa. Deconvolved confocal images, scale bar 10 µm, 2.5 µm for inset. (**b**) 3D-SIM of stable U2OS Cherry-HPOS cells treated and stained as in (**a**) including Alexa 488 Phalloidin to visualize actin. Scale bar 10 µm, 0.5 µm for insets. (**c**) Electron micrographs of U2OS cells treated with 400 µM oleic acid overnight, immunostained with anti-NMIIa antibodies. Arrows indicate NMIIa localisation, left panel: stress fibres, middle panel: LD surface, right panel: between contacting LDs. Scale bar 200 nm (**d**) Live cell Airyscan imaging of U2OS cells transfected with GFP-NMIIa for 24 h and loaded with 400 µM oleic acid overnight. Frame rate 633 ms, scale bar 1 µm. Arrows indicate transient GFP-NMIIa accumulation between dissociating LDs (arrowheads). (**e**) Intensity line profiles to visualize transient GFP-NMIIa accumulation between dissociating LDs at indicated times. This figure is accompanied by Supplementary Fig. 4a,b and Supplementary Movies 1 and 2.

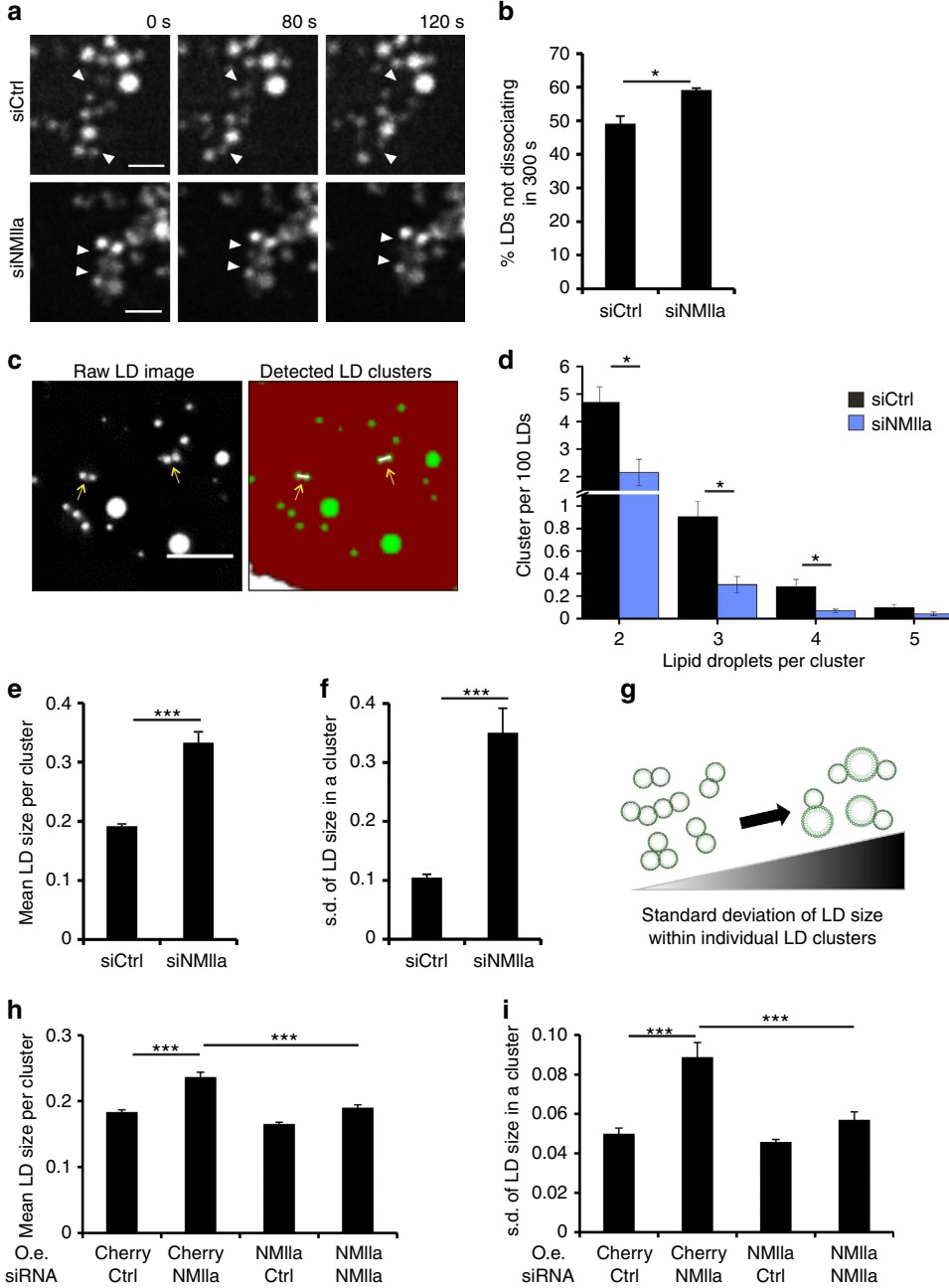

**Figure 3 | Altered clustering and association kinetics of LDs in NMIIa-depleted cells.** (**a**) Live cell CARS microscopy of U2OS cells treated either with siCtrl or siNMIIa, and with oleic acid for overnight. Images were acquired every 2 s for 5 min (scale bar 2 μm for upper panel, 2.5 μm for lower panel). Arrowheads indicate alternating LD associations (upper panel) and stable LD associations (lower panel). (**b**) Quantification of LDs that stay associated during 5 min, for videos acquired as in **a**, oleic acid load was for 8 h to overnight. ± s.e.m., n = 3 independent experiments, at least 290 LD associations were quantified per treatment. (**c**) Example of automated LD cluster detection (scale bar 2.5 μm). Arrows indicate detected LD clusters. (**d**) Automated LD cluster analysis of U2OS cells treated with siCtrl or siNMIIa, and oleic acid for 7 h. Cluster analysis was performed on the same image sets as in Fig. 1d. Number of clusters detected per 100 LDs per cell were quantified for cluster sizes containing 2, 3, 4 and 5 LDs. ± s.e.m. n = 3 independent experiments. (**e**) Automated quantification of mean LD size in clusters. (**f**) Automated quantification of LD size heterogeneity in clusters (for **e** and **f**, ± s.e.m. n = 258 cells for siCtrl and 134 cells for siNMIIa, from three independent experiments). (**g**) Schematic representation for the quantification of LD size heterogeneity within clusters. (**h**) Quantification of mean LD size in clusters in U2OS cells stably overexpressing (o.e.) Cherry or Cherry-NMIIa-siRes treated with siCtrl or siNMIIa, and oleic acid for 8 h. Quantification was performed from the same data set as in Supplementary Fig. 2e (± s.e.m., n = 224 cells for siCtrl U2OS Cherry, 160 for siNMIIa U2OS Cherry, 206 for siCtrl U2OS Cherry-NMIIa siRes and 151 for siNMIIa U2OS Cherry-NMIIa siRes). (**i**) Quantification of LD size heterogeneity from the same data sets as in **h**. Student's t-test *P < 0.05, ***P < 0.001.

Fig. 7a–e). Of note, despite the clear reduction of NMIIa, actin was not reduced in LD fractions of FMNL1-depleted cells (Supplementary Fig. 7h), apparently reflecting the association of also other actin populations, that do not contain NMIIa, to

LDs. Finally, FMNL1 depletion resulted in the formation of enlarged LDs analogously to NMIIa depletion (Supplementary Fig. 8a). This was observed despite the fact that FMNL1 depletion also inhibited fatty acid uptake, thereby lowering

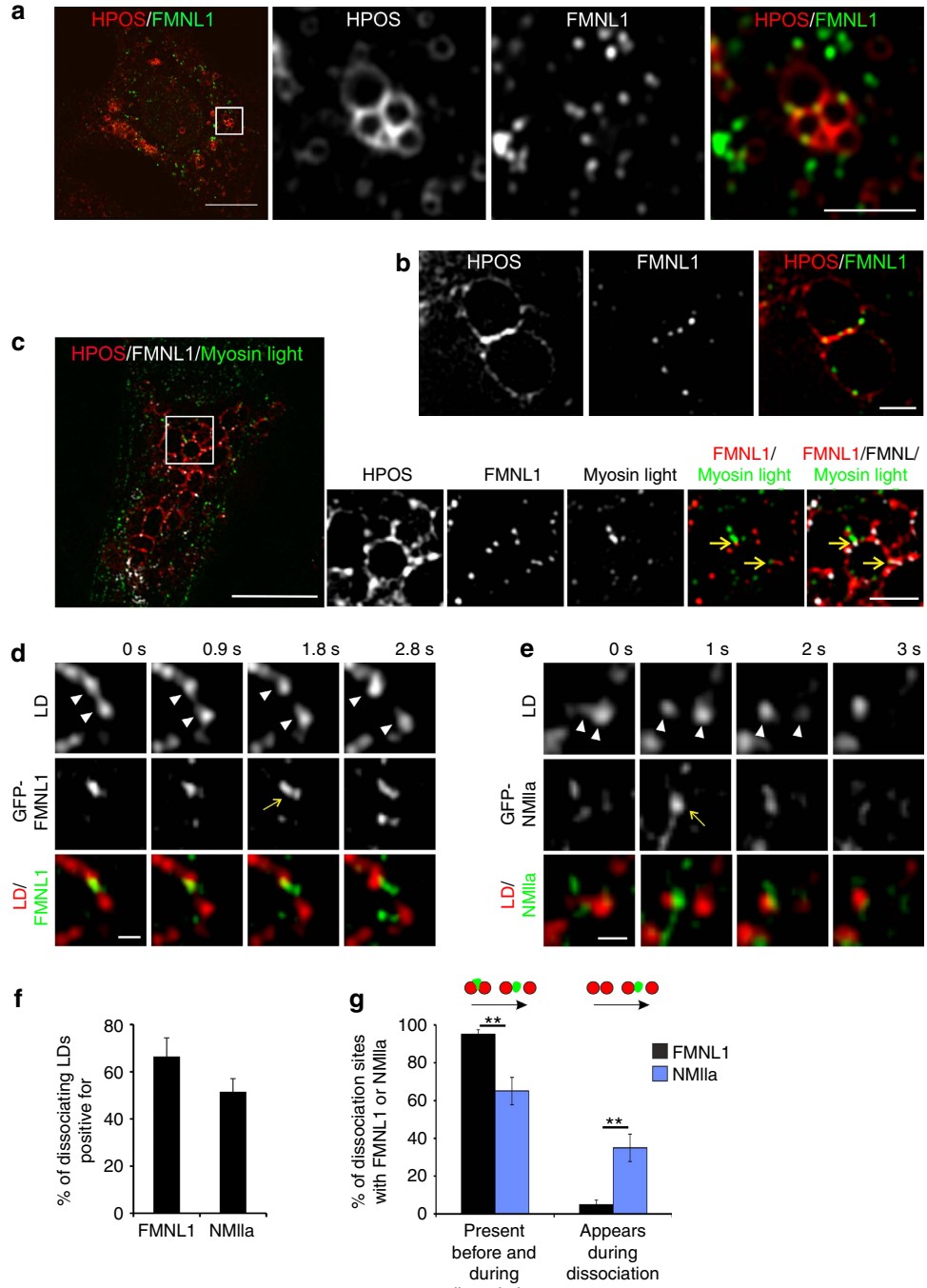

**Figure 4 | FMNL1 localizes to LDs and coincides with LD dissociation sites.** (**a**) Airyscan microscopy of stable U2OS Cherry-HPOS cells treated with 400 μM oleic acid overnight, saponin permeabilized, fixed and immunostained for FMNL1. Scale bar 10 μm, 2.5 μm for inset. (**b**) 3D-SIM of stable U2OS Cherry-HPOS cells treated and stained as in (**a**). Scale bar 1 μm. (**c**) 3D-SIM of stable U2OS Cherry-HPOS cells treated with 400 μM oleic acid overnight, saponin permeabilized, fixed and immunostained for FMNL1 and myosin light chain. Scale bar 10 μm, 1 μm for inset. Arrows indicate close proximity of FMNL1 and myosin light chain on LD surface. (**d**) Live cell Airyscan microscopy of GFP-FMNL1 and LDs in U2OS cells treated with 200 μM oleic acid overnight, images were acquired every 925 ms. Scale bar 0.5 μm. Arrow indicates GFP-FMNL1 accumulation between dissociating LDs (arrowheads). (**e**) Live cell Airyscan microscopy of GFP-NMIIa and LDs in U2OS cells treated with 400 μM oleic acid overnight, images were acquired every second. Scale bar 0.5 μm. Arrow indicates GFP-NMIIa accumulation between dissociating LDs (arrowheads). (**f**) Quantification of LD dissociation events positive for FMNL1 or NMIIa from videos acquired as in (**d,e**). 155 LD dissociation events were assessed from 7 GFP-FMNL1/LD videos and 136 LD dissociation events from 13 GFP-NMIIa/LD videos. (**g**) Timing of FMNL1 and NMIIa recruitment to LD dissociation sites. LD dissociation sites positive for FMNL1 or NMIIa were assessed for whether FMNL1 or NMIIa associated with the site before or only during the dissociation event. The same datasets as in (**f**) was used for quantification. Student's t-test **P < 0.01. See also Supplementary Movies 8 and 9.

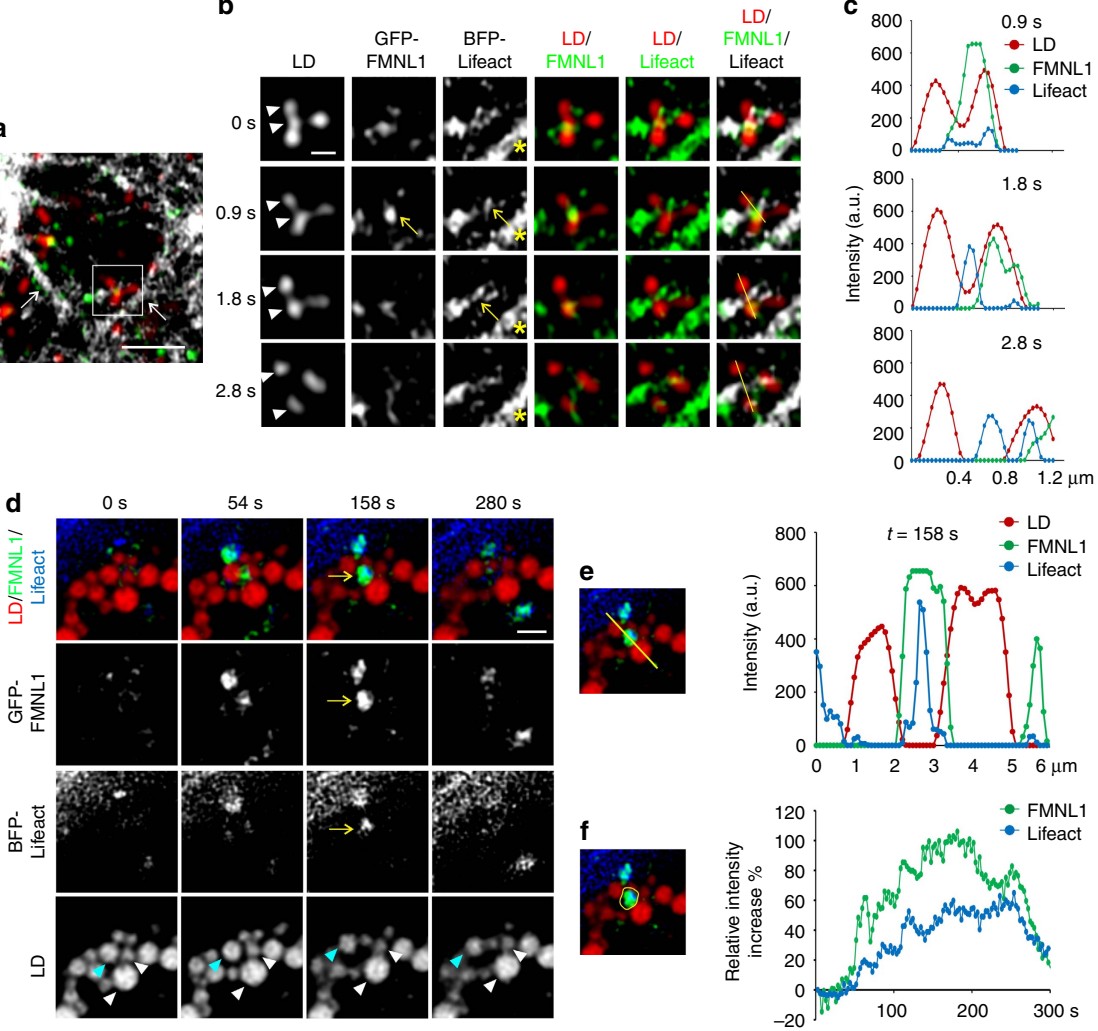

**Figure 5 | Actin and FMNL1 dynamics upon LD dissociation.** Live cell Airyscan microscopy of GFP-FMNL1, BFP-Lifeact and LDs in U2OS cells treated with 200 μM oleic acid overnight, images were acquired every 925 ms (**a–c**). (**a**) An area below the nucleus with prominent ventral stress fibres (white arrows) was imaged, the region shown in (**b**) is indicated. Scale bar 2 μm. (**b**) Visualisation of a LD dissociation event with FMNL1 and Lifeact at the dissociation site. Lifeact also highlights the more stable stress fibre (asterisk). Scale bar 0.5 μm. Arrows indicate transient GFP-FMNL1/BFP-Lifeact accumulation, arrowheads indicate dissociating LDs. (**c**) Line profiles visualize the transient accumulation of FMNL1 and subsequent accumulation of Lifeact at the dissociation site. (**d**) Live cell imaging of GFP-FMNL1, BFP-Lifeact and LDs in U2OS cells treated with 400 μM oleic acid overnight. Cells were switched to growth medium for video acquisition, images were acquired every 2.59 s. Scale bar 2 μm. Arrows indicate transient accumulation of GFP-FMNL1 and BFP-Lifeact between LDs. Arrowheads highlight dissociating LDs with light blue indicating the LD moving most relative to the others. (**e**) Line profiles demonstrate the accumulation of GFP-FMNL1 and BFP-Lifeact between dissociating LDs. (**f**) Intensity measurements of GFP-FMNL1 and BFP-LifeAct between LDs throughout the video in the area indicated. See also Supplementary Movies 12 and 13.

TG storage (Supplementary Fig. 8b,c). Together, these data suggest that FMNL1 mediates LD localisation of NMIIa, possibly by generating specific actin filaments to which NMIIa binds.

To provide more direct evidence for actin filament assembly on LDs and the potential role of FMNL1 in this process, we designed an *in vitro* assay, which allows the monitoring of actin polymerisation on isolated LDs using total internal reflection fluorescence (TIRF) microscopy. LDs were isolated from oleic acid-loaded U2OS cells, labelled with LipidTOX green, and then mixed with rhodamine actin and unlabelled actin in a TIRF reaction chamber (Fig. 6d). Actin filament assembly was monitored by video TIRF microscopy (Fig. 6d,e). We found that under these conditions ∼6% of isolated LDs associated with actin filaments and displayed time-dependent growth of actin filaments from them (Fig. 6e–g). That only a small fraction

of LDs displayed actin nucleation may result from partial dissociation of FMNL1 from LDs during isolation or absence of soluble FMNL1-interacting proteins in the *in vitro* system. Thus, the process may be more efficient in living cells. Addition of the formin inhibitor SMIFH2 (ref. 22) or use of LDs isolated from FMNL1-depleted cells in the *in vitro* assay blunted actin filament association with LDs (Fig. 6g).

**MYH9RD patient neutrophils display altered lipid storage.** Mutations in the human MYH9 gene cause a rare autosomal dominant disorder generally referred to as MYH9-related disease (MYH9-RD), with tens of individual disease causing mutations identified[23]. MYH9-RD is characterized by NMIIa inclusion bodies in neutrophilic leucocytes, thrombocytopenia with giant platelets and multiorgan symptoms, including

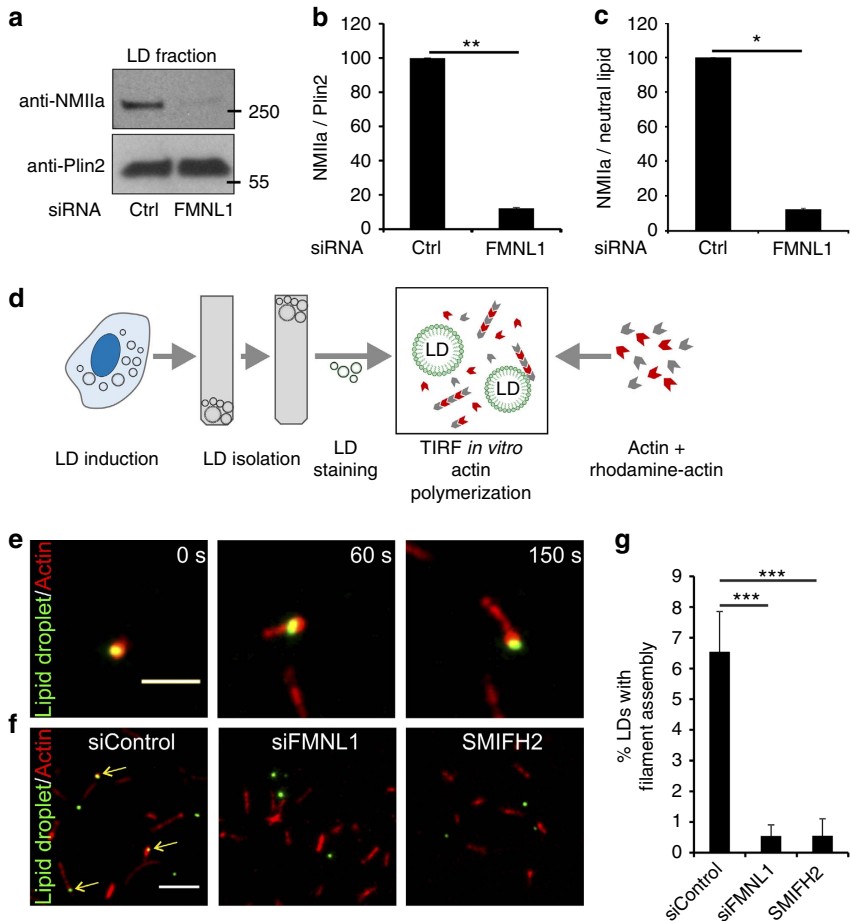

**Figure 6 | FMNL1 regulates NMIIa recruitment to LDs and actin assembly on isolated LDs.** (**a**) FMNL1 depletion reduces NMIIa in LD fractions. Stable U2OS Cherry-HPOS cells were treated with siControl or siFMNL1 and with 400 μM oleic acid overnight. Western blots for NMIIa and Perilipin-2 (Plin2) are shown. (**b,c**) Quantification of NMIIa abundance in LD fractions relative to Plin2 (**b**) or neutral lipid content (**c**), ± s.d. from two independent experiments. (**d**) Schematic representation of TIRF microscopy based monitoring of actin assembly on isolated LDs. LDs were isolated by density gradient centrifugation from U2OS cells treated with siCtrl or siFMNL1 and loaded with 400 μM oleic acid overnight. A mix of 0.15 μM unlabelled non-muscle actin, 0.065 μM Rhodamine non-muscle actin and 0.43 μM profilin was polymerized in the presence of LDs. Actin polymerisation was imaged by time-lapse TIRF microscopy for 15 min at 100 ms intervals. (**e**) Representative example of actin filament assembly on an isolated LD. Scale bar 2.5 μm. (**f**) Representative images from actin assembly studies with LDs isolated from control cells (left image), siFMNL1-treated cells (centre image) or LDs from control cells incubated with formin inhibitor (SMIFH2) (right image). Scale bar 5 μm. Arrows indicate LDs with associated actin filaments. (**g**) Quantification of nascent actin filaments associated with LDs. LDs stably associated with actin filaments were quantified and normalized to the total LD number per image field. ± s.e.m., $n = 11$ assays for siCtrl, 11 assays for siFMNL1 and 7 assays for formin inhibition. Student's t-test *$P < 0.05$, **$P < 0.01$, ***$P < 0.001$.

sensorineural hearing loss, renal failure and presenile cataract, the pathogenesis of which are incompletely understood[24]. To investigate whether cells from patients with MYH9-RD display a LD-related phenotype, we isolated peripheral blood neutrophilic leucocytes from two patients with MYH9-RD carrying a point mutation in the C-terminal non-helical tail of NMIIa (S. Ryhänen, personal communication) and two healthy individuals. These patients present with marked macrothrombocytopenia and several family members have been diagnosed with hearing loss[25], in accordance with the site of the mutation[23]. By NMIIa immunostaining, the diagnostic NMIIa inclusion bodies of the disease were evident and the overall NMIIa immunoreactivity was decreased in the patient neutrophils (Fig. 7a,b). Under basal conditions (fasting blood samples, no lipid loading *in vitro*), both control and patient neutrophils displayed few LDs (approximately seven LDs per cell) (Fig. 7c,d). Remarkably, upon 1 h of oleic acid loading *in vitro*, LD accumulation was more pronounced in the patients' neutrophils, resulting in roughly twofold more LDs compared with controls

(Fig. 7d,e). In parallel, there was a tendency for the enlargement of LDs in the patients' cells, especially in the bigger LD size classes (Fig. 7e). Thus, the LD phenotypes in the patient neutrophils appeared partly reminiscent of those observed in NMIIa-silenced U2OS cells. Together, these findings provide the first evidence that NMIIa dysfunction in man causes a lipid imbalance, possibly contributing to the pathogenesis of MYH9-RD. For instance, neutrophil LDs can modulate the immune response, as impaired LD lipolysis was found to be accompanied by reduced lipid mediator release[26].

## Discussion

Actin-binding proteins are prominent constituents of the LD proteome but their specific roles on LDs are not well understood. In this study, we describe a functional interplay of the actin-polymerising factor FMNL1 and NMIIa, a force generating protein, in LD dynamics. NMIIa is generally known as a major constituent of contractile stress fibres, regulating cell

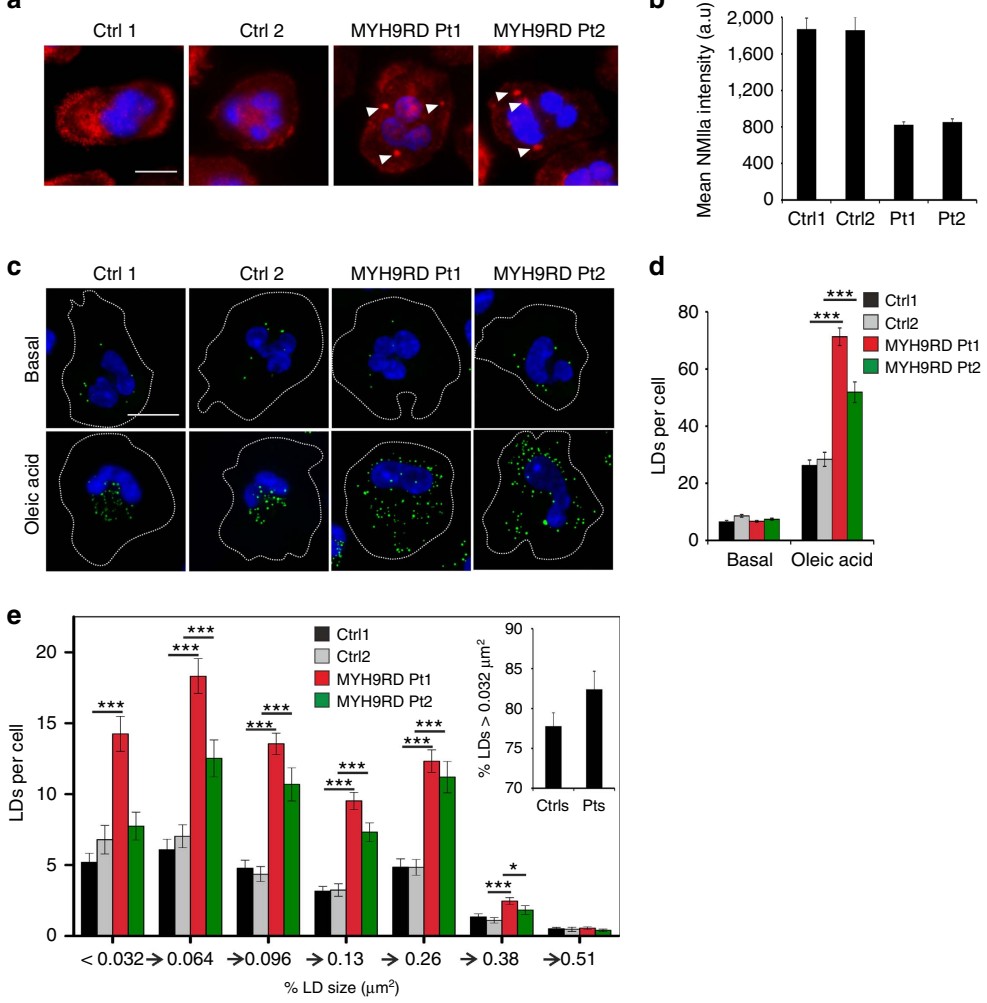

**Figure 7 | Effect of MYH9-RD on LDs in primary human neutrophils.** (**a**) Visualisation of NMIIa inclusions (indicated by arrowheads) in MYH9-RD patients (Pt1, Pt2) neutrophils. Scale bar 5 µm. Image intensities were adjusted to ensure visibility of NMIIa inclusions in patient cells. (**b**) Quantification of NMIIa intensity in control and patient neutrophils. ± s.e.m. $n = 6$–22 cells for each. (**c**) LD staining (LD540) of control and patient neutrophils in basal conditions and upon loading with 10 µM oleic acid for 1 h. Scale bar 10 µm. (**d**) Quantification of LD numbers in (**c**). ± s.e.m. $n = 40$ cells for Ctrl 1; 38 cells for Ctrl 2; 54 cells for Pt 1; 35 cells for Pt 2. (**e**) Size distribution of LDs upon oleic acid loading, same data set as in (**d**). The inset displays the relative amount of LDs larger than 0.032 µm$^2$ ± s.d. Student's t-test, *$P < 0.05$, ***$P < 0.001$.

migration and focal adhesion stability. NMIIa binds actin filaments and ATP hydrolysis evokes a conformational change, resulting in pulling along actin filaments and force generation[27]. Recent evidence indicates that NMIIa can also act on specialized actin filament structures, for instance, during mitochondrial fission[28].

We found that NMIIa transiently accumulates at sites where LDs separate from one another. LDs are known to undergo homotypic associations and dissociations, resulting in the formation of dynamic LD clusters. Our data show that NMIIa has a major impact on cellular LDs, with NMIIa depletion resulting in enlarged LDs and increased TG storage owing to impaired lipid breakdown. We postulate that impaired dissociation/prolonged association of LDs enhances their coalescence, either via direct fusion (Supplementary Fig. 4e,f; Supplementary Movie 7) and/or via lipid transfer[5]. This would result in an increased size heterogeneity of LDs and a net consumption of LDs within a cluster, with a concomitant reduction of clusters. Indeed, NMIIa silencing reduced LD dissociations. In parallel, the LD clustering propensities were altered: there were fewer LD clusters and in these big

LDs often associated with small ones. This is strikingly different from control cells where similarly sized LDs formed clusters.

NMIIa requires a preformed actin filament to exert its function. But how could an actin filament be generated at LD dissociation sites? We demonstrate that FMNL1, a protein with actin polymerisation activity, localizes to LD dissociation sites. It can thus provide a seed for acto-myosin assembly, as depletion of FMNL1 abolishes NMIIa association with LDs. Furthermore, we found that isolated LDs have the ability to induce actin filament assembly *in vitro*. This activity was dependent on FMNL1, as pharmacological inhibition of formins or depletion of FMNL1-blunted actin filament assembly on isolated LDs. Based on these observations, we suggest that FMNL1 localizes to LD dissociation sites to nucleate actin filaments that are functionalized by NMIIa, providing a force to assist in LD dissociation. This idea is supported by the temporal association of the actin-binding proteins with LDs, with FMNL1 accumulation typically occurring between closely apposed LDs before dissociation, potentially defining LD dissociation sites. Although most dissociating LDs also harbour NMIIa, it may also be recruited to the site once LDs are moving apart.

Considering that LDs also undergo rapid reassociations, an alternative, although not mutually exclusive explanation is that FMNL1, in concert with actin and NMIIa, prevents the reassociation and fusion of LDs. Both mechanisms, promotion of LD dissociation and prevention of LD reassociation, highlight the importance of LD dynamics in controlling the coalescence and surface to core ratio of LDs, and thereby the accessibility of modifying enzymes to core TGs.

## Methods

**Antibodies, dyes and reagents.** Rabbit anti-NMIIa (Covance, PRB-440P Immunofluorescence (IF) 1:500, Immunoblot (IB) 1:3,000), mouse anti-alpha tubulin (Sigma Aldrich, T6074, IB 1:20 000), rabbit anti-FMNL1 (Sigma Aldrich, HPA008129, IB 1:1,000), rabbit anti-FMNL1 (D. Billadeau, Mayo Clinic, Rochester, Minnesota, IF 1:50)[21], mouse anti-myosin light chains (Sigma Aldrich, M4401, IF 1:200), Alexa488 goat anti-rabbit, A11008; Alexa568 goat anti-rabbit, A11011; Alexa647 donkey anti-rabbit, A31573 (all Thermo Fisher, 1:200); HCS LipidTOX green neutral lipid stain, H34475, (live cell 1:1,000, fixed cells 1:100) HCS LipidTOX deep red neutral lipid stain, H34477, (live cell 1:5,000 − 1:10,000), Alexa488 Phalloidin, A12379, (132 nM) all Thermo Fisher, LD540 (Princeton BioMolecular Research, 0,1 µg ml$^{-1}$); Cell culture reagents and general reagents were purchased from GibCo/Thermo Fisher, Lonza and Sigma-Aldrich. Lipoprotein-deficient serum has been made from fetal bovine serum (FBS) as described previously[29].

**siRNAs, plasmids and mutagenesis.** SiRNAs targeting NMIIA and FMNL1 were purchased from Qiagen and control siRNA from Sigma-Aldrich. Target sequence siControl: 5′-CGTACGCGGAATACTTCGA-3′ (ref. 30); siNMIIa: 5′-TGGGAGC ACAAGCGCAAGAAA-3′; siNMIIa-2: 5′-CACGGAGATGGAGGACCTTAT-3′, siFMNL1: 5′-TCCGCTGTGGCCCGCCTCAAA-3′, siTM4: 5′-AATTAAACT TCTGTCTGACAA-3′ (ref. 17), eGFP-C3-NMIIa was reported previously[31] (Addgene ID 11348). To generate mCherry-C3-NMIIa, mCherry was first PCR amplified from pmCherry-C1 using the following oligonucleotides CherryFW: 5′-GAGAGCTAGCGCTACCGGTCGC-3′ and CherryBW: 5′-TCCTAGATC TGAGT-ACTTGTACAGCTCGTCCATGC-3′, and used to replace eGFP in eGFP-C3 via NheI and BglII restriction sites to generate a pmCherry-C3 vector. NMIIa was shuttled from eGFP-C3-NMIIa to pmCherry-C3 using HindIII and SalI sites. For generation of siRNA-resistant NMIIa we first generated two PCR fragments including the siRNA resistant sequence in the corresponding -3′ and 5′-oligonucleotides. PCR oligonucleotides for 1st fragment: NMIIa5Prime: 5′-GAG AAAGCTTATGGC ACAGCAAGCTG-3′; 3Prime-siRNA-NMIIA: 5′-CTTTTTCC GTTTATGTTCGGAGTCCCCTTTGCCCTGCAGCAG-3′, PCR oligonucleotides for the 2nd fragment: 5Prime-siRNA-NMIIa: 5′-TCCGAACATAAACGGAAA AAGGTGGAGGCCGAGGG-3′ and NMIIa-3Prime-EcoR1: 5′-GAGAGAAT TCTTCTCGTCCTCCACCTG-3′. The fragments were used as template for amplification of an siRNA resistant N-terminal NMIIA fragment using NMIIa5Prime and NMIIa-3Prime-EcoRI oligonucleotides. This was inserted into GFP-NMIIa via HindIII and EcoRI restriction sites to generate GFP-NMIIa-siRes. NMIIa siRes was transferred from GFP-NMIIa to mCherry-C3 to generate mCherry-NMIIa-siRes. For the generation of Cherry-HPOS, oligonucleotides encoding the HPOS sequence[18] with a 5′ BspEI restriction site 5′-TCCGG AATGGATGTCCTGGTTCCATTGCTGCAGCTGCTGGTGCTGCTCCTGAC TCTGCCTTTACAC-CTGCTGGCTCTGCTGGGCTGCTGGCAGCCCCTCTT TGAAGCGATTGGCAAGATATTCAGCAATATCC-GCATCAGCACGCAGA AAGAGATATGA-3′ were synthesized, annealed and inserted into pmCherry-C1 via BspEI and SmaI restriction sites. We obtained the cDNA for FMNL1 alpha from D. Billadeau[21], PCR amplified it with the oligonucleotides: FMNL1FWNhe1: 5′-GAGAGCTAGCATGGGCAACGCGGCCGGCA-3′ and FMNL1BWBamH1: 5′- GAGAGGATCCCGGAGGGGGCATCTCTTCTCCCA-3′ and cloned into peGFP-N1 via Nhe1 and BamH1 restriction sites. BFP-actin was generated by replacing GFP of GFP-actin with mTagBFP2 from pmTagBFP2-C1 (from V. Verkhusha) using Nhe1 and Bgl2 restriction sites. BFP-LifeAct was obtained by replacing the GFP-sequence of LifeAct-GFP with mTagBFP2 from pmTagBFP2-N (from V. Verkhusha) via BamH1 and Not1 restriction sites.

**Cell culture and transfection.** U2OS cells kindly provided by Marikki Laiho (John Hopkins University, USA) were cultured in Dulbecco's Modified Eagle's Medium (DMEM) with 15% FBS, supplemented with penicillin/streptomycin (100 U ml$^{-1}$ each) and L-glutamine (2 mM). Stable U2OS Cherry-NMIIa, Cherry-NMIIa-siRes and Cherry-HPOS cell lines were selected with 500–600 µg G418 per ml culture medium. SiRNA treatments were performed for 48 or 72 h using Hiperfect (Qiagen) or RNAiMax (Thermo Fisher) using reverse transfection with 50 nM siRNA. Oleic acid loading was performed during the siRNA treatments. Plasmids were transfected using Lipofectamine LTX (Thermo Fisher) with the PLUS (Thermo Fisher) reagent. Cells were tested for Mycoplasma infection.

**LD isolation and mass spectrometry.** Raw 264.7 cells (received from Vesa Olkkonen, Minerva Foundation Institute for Medical Research, Helsinki, Finland) were cultured in DMEM with 10% FBS, 10 mM 4-(2-hydroxyethyl)-1-piper-azineethanesulfonic acid (HEPES), L-glutamine (2 mM) and penicillin/strepto-mycin (100 U ml$^{-1}$). Cells were treated with 400 µM oleic acid or 35 µg ml$^{-1}$ acetylated low density lipoprotein in DMEM with 5% lipoprotein deficient serum for 24 h. LD isolation was performed similar to previously described[32]. The treatments were combined for LD isolation to maximize the yield. Cells were washed with cold phosphate buffered saline (PBS) and resuspended in 3.5 ml hypotonic lysis buffer (HLM; 20 mM Tris pH 7.4, 1 m M ethylenediaminetetraacetic acid), and incubated on ice for 10 min. Cells were disrupted by repetitive passaging through a 25 g needle, 3 ml of cell suspension was transferred to a Beckmann polyallomer tube (14 × 95 mm) (No. 331374) and mixed with 1.5 ml 60% sucrose solution, overlayed with 4 ml 5% sucrose solution and 4 ml HLM Buffer. Samples were centrifuged at 28,000 g with a SW40 rotor for 2 h. The LD fraction was recovered by tube slicing and subjected to liquid chromatography–mass spectrometry (LC-MS/MS). For LC-MS/MS, the samples were prepared as follows: the cysteine bonds were reduced with 5 mM TCEP (Sigma–Aldrich), alkylated with 10 mM iodoacetamide (Sigma–Aldrich), and the proteins were digested to peptides by trypsin (Promega, Madison, WI). After overnight incubation at 37 °C, samples were quenched with 10% TFA, purified with C18 Micro SpinColumns (Harvard Apparatus, Holliston, MA) and re-dissolved in 30 µl 0.1% TFA, 1% CH3CN. The LC-MS/MS analysis was carried out on an EASY-nLCII nanoflow system (Thermo Scientific) connected to a Velos Pro-Orbitrap Elite hybrid mass spectrometer (Thermo Scientific) using the Xcalibur version 2.7.1. The peptides were separated using 10 cm analytical column (Thermo Fisher Scientific) with 60 min gradient ranging from 5 to 35% buffer B (0.1% formic acid in 98% acetonitrile and 2% water), followed by a 5 min gradient from 35–80% buffer B and 10 min gradient from 80–100% buffer B at a flow rate of 300 nl min$^{-1}$. Proteome Discoverer software (Thermo Scientific) together with SEQUEST search engine was used for peak extraction and subsequent peptide and protein identification. Error tolerances on the precursor and fragment ions were 15 ppm and 0.6 Da, respectively. Database searches were limited to fully tryptic peptides with maximum one missed cleavage. For peptide identification the FDR was set to <5%.

**LD isolation and actin assembly on isolated LDs.** U2OS cells (6–7 10 cm dishes) were treated overnight with 400 µM oleic acid in culture medium. EDTA in HLM buffer was replaced with EGTA. LD isolation was performed as described above and LD fractions were retrieved from the top with a 1 ml pipette. In vitro TIRF imaging was performed as previously described[33], but muscle actin was substituted with non-muscle actin (Cytoskeleton Inc.), prepared according to the manufacturer's instructions, and non-muscle rhodamine actin (Cytoskeleton Inc.) was used for labelling of the filaments. Videos were obtained with 0.15 µM unlabelled non-muscle actin, 0.065 µM rhodamine non-muscle actin and 0.43 µM profilin concentrations. Profilin was purified from cow spleen as previously described[34] SMIFH2 formin inhibitor[22] (Sigma-Aldrich) was used at 100 µM. TIRF imaging was performed in mPEG-Silane (MW 5k)-coated chambers with a Nikon Eclipse Ti-E N-STORM microscope, equipped with Andor iXon+ 885 EMCCD camera and × 100 Apo TIRF oil objective NA 1.49, a 561 nm laser line was used for visualisation of Rhodamine actin. Actin filament polymerisation was followed in 0.1 s intervals for 15 min after addition of the reaction into the imaging chamber.

**LD preparation for western blot analysis.** U2OS Cherry-HPOS cells were seeded in 2–4 10 cm dishes and transfected with siControl or siFMNL1. After 24 h cells were incubated with 400 µM oleic acid in control medium for an additional 24 h. Cells were washed 2 × with Dulbecco's phosphate-buffered saline (DPBS) and then scraped into 5 ml DPBS. Cells were pelleted by centrifugation at 730 g for 5 min and resuspended in 900 µl HLM buffer. Cells were broken by repetitive passaging through a 25 g needle, and 750 µl of the suspension was mixed with 750 µl HLM buffer with 60% sucrose, overlayed with 1 ml HLM with 5% sucrose followed by 1.5 ml HLM buffer. LDs were floated by centrifugation at 28,000 g for 2 h in a SW60 rotor. Of the LD fraction, 10% was removed for lipid determination and the rest was subjected to acetone precipitation with 10 × volume of − 75 °C acetone. Samples were incubated at − 20 °C degree overnight and centrifuged at 3,300 g for 100 min. Proteins were resuspended in 2 × loading buffer by sonication and subjected to western blot analysis. Original scans of the respective western blots shown in Fig. 6a can be found in Supplementary Fig. 9.

**Automated LD analysis.** U2OS cells were cultured on coverslips or 96 well Screenstar plates (Greiner) and incubated with or without oleic acid for 7 h. Cells were treated with 2 µM CellTracker Red for 30 min to 1 h in culture medium with or without oleic acid, washed 3 × with PBS and fixed with 4% paraformaldehyde in 250 mM HEPES, 1000 µM CaCl, 100 µM MgCl, pH 7.4. Cells were washed 3 × with PBS and stained with 1 µM 4′,6-diamidino-2-phenylindole (DAPI) and 100 ng ml$^{-1}$ LD540 (or 1:1,000 LipidTox green) in PBS for 30 min at room temperature (RT), washed 3 × with PBS and mounted with PBS and epoxy glue. For every treatment 10 image stacks were acquired for DAPI, LD and cell tracker

channels using a Nikon Eclipse Ti-E inverted microscope equipped with a 60 × PlanApo VC oil objective NA 1.40 (or 40 × Planfluor objective with NA 0.75 and 1.5 zoom). Image stacks were automatically deconvolved using the Huygens batch processing application (Scientific Volume Imaging) and deconvolved image stacks were transformed into maximum projection images using custom MatLab scripts. Cell segmentation and LD detection was performed with CellProfiler 1.0.5122 (CellProfiler.org)[35] in a hierarchical manner. First, cell nuclei with a typical diameter between 100 and 250 pixels were detected in DAPI images based on the Otsu adaptive thresholding method. Touching nuclei were separated by build in intensity methods. Second, the cytoplasm was detected in CellTracker Red images using intensity propagation based on the Otsu global thresholding method using the identified nuclei from the first step as a seed point. Third, LDs were detected using a combination of a A-trous spot detection method with Otsu adaptive thresholding, enabling the detection of small and large LDs separately. The detected LDs were further processed with custom Matlab scripts, which corrected for large LDs misclassified as several small LDs. Furthermore, the custom Matlab scripts contained an algorithm to identify clustered LDs. Data visualisation was done using Excel and Origin 8.6.

**Lipid extraction, analysis and TG hydrolysis.** Lipid extraction was performed as described previously[36]. Cells were washed 3 × with PBS and scraped into 2% sodium chloride and lipids were extracted with chloroform methanol in a 2:1 ratio. Solvents were evaporated and lipids dissolved in 2:1 chloroform: methanol for application to thin layer chromatography (TLC). TLC was performed with hexane:diethylether:acetic acid 80:20:1. TLC plates were stationed with copper sulfate and developed by charring. TG content was quantified and normalized to protein content measured by Bio-Rad protein determination. For the TG hydrolysis assay, siCtrl- and siNMIIa-treated U2OS cells were loaded with 200 μM oleic acid for 24 h, with or without inhibition of cholesterol ester synthesis with Sandoz-58-035. Cells were washed with PBS and incubated with U2OS growth medium for 8 h. Lipids were extracted before or after the 8 h chase in normal growth medium. TG content was quantified and used to calculate the percent decrease in TG before the 8 h chase. In cells treated with control siRNA TGs were reduced on average by 45% during the 8 h chase.

**TG synthesis and fatty-acid uptake.** U2OS cells were treated with NMIIa or control siRNA for 48 h. Cells were washed 3 × with PBS and then treated with [3]H-oleic acid (Perkin Elmer) in DMEM plus 2% fatty acid-free bovine serum albumin for 2 h at 37 °C. Cells were washed 3 × with PBS, lipids were extracted and neutral lipids resolved by TLC. TGs were scraped into scintillation solution and radioactivity was measured. Counts were normalized to an internal standard and protein content. For fatty-acid uptake, experiments cells were treated with [3]H-oleic acid in U2OS growth medium for 15 min. Cells were washed 3 × with PBS and cell extracts were made in 2% NaCl. Radioactivity of the cell lysate was measured with scintillation counting and normalized to protein content.

**Western blot analysis.** Cells were washed 3 × with PBS and lysed with Tris-buffered saline (TBS)/1% Triton X-100. Lysates were incubated on ice for 10 min, vortexed and centrifuged for 10 min at 16,000 g at 4 °C. Cleared supernatants were mixed with loading buffer (50 mM Tris pH 6.8, 2% sodium dodecyl sulphate (SDS), 1% β-mercaptoethanol, 5% glycerol, final concentrations) and equal protein amounts were separated on SDS-polyacrylamide gels and transferred to Immobilon polyvinylidene fluoride (Millipore) membranes. For Odyssey antibody detection, blots were blocked with Odyssey Blocking buffer/PBS (1:1) overnight at RT. Primary antibodies were diluted in Odyssey Blocking Buffer/PBS with 0.1% Tween and incubated for 2 h at RT. Blots were washed with TBS/0.1% Tween and incubated with secondary antibodies in Odyssey Blocking Buffer/PBS with 0.1% Tween and 0.01% SDS for 30 min at RT. Blots were washed with TBS/0.1% Tween and PBS and scanned with an Odyssey CLx detection system. For enhanced chemiluminescence (ECL) detection blots were blocked with TBS/0.1% Tween with 5% skim milk for 2 h at RT. Primary antibodies were diluted in blocking buffer and incubated overnight at 4 °C. Blots were washed with TBS/0.1% Tween and incubated with secondary antibodies in blocking buffer for 30 min. Blots were washed with TBS/0.1% Tween, PBS and distilled water. ECL was performed with Clarity Western ECL substrate (Bio-Rad).

**Quantitative PCR.** RNA was purified from cells using a Nucleospin RNA isolation kit (Macherey-Nagel) following the manufacturer's protocol and reverse transcribed with a SuperScript VILO cDNA synthesis kit (Thermo Fisher Scientific). Real-time PCR was performed with Lightclycer 480 SYBR Green master mix (Roche Diagnostics), 250 nM oligonucleotides and 64 °C annealing temperature. GAPDH FW: 5′-GAAGGTGAAGGTCGGAGTC-3′ GAPDH BW: 5′-GAAGATG GTGATGGGATTTC-3′; FMNL1 QFW1: 5′-CTCAGGAGGCCTTTGAGTCTG-3′; FMNL1 QBW1: 5′-ACCTCCTGCTCAGCTTTCTTGTA-3′.

**IF staining using pre-fixation permeabilization.** U2OS Cherry-HPOS cells were seeded on fibronectin coated cover slips and incubated overnight with 400 μM oleic acid. Cells were washed with DPBS, permeabilized with 0.005% saponin in cyto-buffer (10 mM MES pH 6.1, 138 mM KCl, 3 mM MgCl$_2$, 2 mM EGTA and 320 mM sucrose) for 1 min, washed once with cyto-buffer and fixed with cytoskeleton preservation fix (2.6% paraformaldehyde in 160 mM HEPES, 64 μM CaCl$_2$, 67 μM MgCl$_2$, 10 mM MES pH 6.1, 138 mM KCl, 2 mM EGTA and 320 mM sucrose) for 30 min. Cells were washed with cyto-buffer, quenched with 50 mM NH$_4$Cl in cyto-buffer for 5 min, washed with cyto-buffer and blocked with 0.2% BSA in cyto-buffer (blocking buffer) for 10 min at RT. Primary antibody incubations were done in blocking buffer for 1 h at RT, followed by three washes with cyto-buffer for 5 min. Secondary antibody was diluted in blocking buffer and incubated for 45 min at RT. Cell were washed 3 × 5 min with cyto-buffer and mounted with MOWIOL/DAPCO for confocal or Airyscan microscopy or with ProLong Diamond for 3D-SIM microscopy.

**IF staining using post-fixation permeabilization.** Cells were fixed with 4% paraformaldehyde in 250 mM HEPES, pH 7.4, 100 μM CaCl$_2$ and 100 μM MgCl$_2$ for 15 min or cytoskeleton preservation fix for 15–30 min. Cells were quenched with 50 mM NH$_4$Cl for 10 min and permeabilized with PBS/0.1% Triton X-100 for 5 min. Cells were blocked with 10% FBS in PBS for 30 min at 37 °C. Primary antibodies were diluted in blocking solution and incubated for 45 min at 37 °C. Cells were washed 3 × with PBS and incubated with anti-rabbit Alexa 546 or Alexa 647 in PBS/0.1% Tween in the presence or absence of Alexa 488 Phalloidin for 45 min at 37 °C. Cells were washed 3 × with PBS and mounted with Mowiol/DAPCO.

**Confocal and superresolution light microscopy.** For confocal microscopy, cells were imaged using a Leica TCS CARS SP8 confocal microscope with a 63 × HC PL APO CS2 glycerol objective NA 1.3. or a Zeiss LSM 880 confocal microscope using a 63 × Plan-apochromat oil objective, NA 1.4. For Airyscan superresolution microscopy, cells were imaged with a Zeiss LSM 880 equipped with an Airyscan detector using a 63 × Plan-apochromat oil objective, NA 1.4. The Airyscan detector was adjusted before each image or video acquisition and images were Airyscan processed using the Zeiss Zen software package. 3D-SIM was performed on a Deltavision OMX V4 microscope (Applied Precision, GE Healthcare) equipped with an Olympus 60x Plan Apochromat objective NA 1.42, 488, 568 and 642 nm laserlines, and three cooled sCMOS cameras. z-Stacks covering the whole cell were recorded, with a z-spacing of 125 nm. For each focal plane, 15 raw images (five phases for three different angular orientations of the illumination pattern) were captured. Superresolution images were reconstructed, aligned, and processed for presentation using Softworx software (Applied Precision, GE Healthcare). Brightness and contrast were adjusted with ImageJ, please note that cropped areas may have different intensity adjustments than the corresponding larger cell views. Localisation of NMIIa at LDs was assessed in 11 independent experiments; localisation of NMIIa at LDs upon FMNL1 depletion in three independent experiments; colocalisation of NMIIa and phalloidin at LDs in eight independent experiments; FMNL1 localisation at LDs in eight independent experiments; localisation of FMNL1 and myosin light chain at LDs in three independent experiments; localisation of myosin light chain at LDs in five independent experiments.

**Quantification of NMIIa and FMNL1 localisation at LDs.** Confocal image stacks of U2OS Cherry-HPOS cells stained with anti-FMNL1 or anti-NMIIa antibodies were deconvolved using Huygens deconvolution software. NMIIa and FMNL1 patches with a preset minimum intensity were counted at LDs with a diameter of ≥1 μm by inspecting each individual section of a z-stack. On average, 10 LDs were counted per cell. NMIIa or FMNL1 patches, which associated with two neighbouring LDs were counted as patches at LD contact sites.

**Live cell imaging.** U2OS cells were seeded in LabTEK chamber slides coated with 5 μg human fibronectin (Roche Diagnostics). Cells were treated with 200– 400 μM oleic acid for 8 h or overnight. Cells were imaged in fresh culture medium or culture medium containing oleic acid including 10 mM HEPES pH 7.3. LDs were stained either with LipidTOX green or LipidTOX deep red. Live cell microscopy was performed with a Leica TCS SP8 confocal microscope or a LSM880 confocal microscope with an Airyscan module. Live cell microscopy for fluorescently tagged NMIIa at LDs was performed in nine independent experiments; GFP-FMNL1 at LDs in four independent experiments; GFP-FMNL1 with BFP-Lifeact at LDs in six independent experiments and BFP-Lifeact/BFP-actin at LDs in four independent experiments. LD dynamics with blebbistatin was assessed in four independent experiments and with cytochalasin D in one experiment.

**CARS microscopy.** CARS microscopy is a non-linear label-free imaging method especially well suited for LD imaging[37,38]. CARS video microscopy was done with non-labelled cells by adjusting pump and stokes laser to the vibration frequency of C-H groups (2845 cm$^{-1}$). An infrared corrected 25 × objective (HCX IR APO L, NA 0,95) was used for CARS image acquisition. Images were acquired every 2 s over a time course of 5 min. Live cell CARS microscopy for LD association dynamics was assessed in three independent experiments (Fig. 3a,b, Supplementary Movie 4).

**Pre-embedding immuno-EM.** U2OS cells were loaded with 400 µM oleic acid overnight and fixed with 4% paraformaldehyde in MES-sucrose buffer (10 mM MES/138 mM KCl/3 mM MgCl$_2$/2 mM EGTA/11.14% sucrose) for 2 h. The cells were then permeabilized with 0.01% saponin in 0.1% BSA/0.1 M Na-PO$_4$ pH 7.4 for 8 min, labelled with the rabbit anti-NMIIa antibody (Covance, PRB-440 P 1:50) for 2 h followed by the nano-gold-conjugated anti-rabbit Fab-fragments for 1 h (Nanoprobes, #2004, 1:60), post-fixed in 10% glutaraldehyde, and quenched in 50 mM glycine. The nano-gold particles were intensified using the HQ SILVER Enhancement kit according to the manufacturer's instructions (Nanoprobes) followed by gold toning in subsequent incubations in 2% NaAcetate, HAuCl$_4$ and 0.3% Na$_2$S$_2$O$_3$•5H$_2$O. The cells were treated with 1% OsO$_4$ supplemented with 15 mg ml$^{-1}$ K$_4$[Fe(CN)]$_6$ and flat-embedded in Epon as described earlier[39]. Ultrathin sections were cut using Leica UCT microtome, picked on pioloform coated single slot grids, post-stained with uranyl acetate and lead citrate, and observed with the Jeol JEM-1400 microscope at 80 kV. Images were acquired with the Gatan Orius SC 1000B camera.

**Neutrophil isolation and lipid loading.** Freshly isolated peripheral blood from MYH9-RD patients and from healthy individuals was used for neutrophil isolation using Polymorphprep (Axis-Shield) and contaminating erythrocytes were lysed using 155 mM NH$_4$Cl, 12 mM NaHCO$_3$, 0.1 mM EDTA. Neutrophils were resuspended in Hank's balanced salt solution without calcium and magnesium and adhered to poly-D-lysine-coated cover glasses for 15 min or transferred to 24 well plates with poly-D-lysine-coated cover glasses and incubated with 10 µM oleic acid for 1 h at 37 °C. Neutrophils were fixed with 4% paraformaldehyde and subjected to NMIIa or LD stainings. Neutrophils were imaged with a Nikon Eclipse Ti-E N-STORM microscope, equipped with a 100 × Apo TIRF oil objective NA 1.49. LD images were automatically deconvolved using Huygens software (Scientific Volume Imaging). Maximum intensity projections were generated using MatLAB and thresholded LD images were quantified using the ImageJ analyse particles tool. The MYH9-RD patients provided a written informed consent in accordance with the Declaration of Helsinki.

**Data analysis.** Data are presented as mean ± s.e.m. Statistical significance was calculated with students $t$-test (two tailed) using Microscoft Excel.

**Code availability.** Matlab scripts and corresponding CellProfiler analysis pipelines (CellProfiler 1.0.5122, www.cellprofiler.org) for automated LD analysis can be obtained from Peter Horvath (peter.horvath@brc.mata.hu).

**Data availability.** The data that support the findings of this study are available in the article, including Supplementary Files, or are available from the authors upon reasonable request.

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

## Acknowledgements

We acknowledge technical assistance by Anna Uro and support by the Biomedicum Imaging Unit, University of Helsinki, especially Mikko Liljeström and Antti Isomäki. We thank David Billadeau (Mayo Clinic, Rochester, USA) for providing FMNL1 antibodies and FMNL1 cDNA constructs; Konstantin Kogan (University of Helsinki) for purified profilin, Shiqian Li (University of Helsinki) for Cherry-HPOS plasmid, Vladislav Verkhusha (University of Helsinki) for pmTagBFP2 plasmids. We thank the Electron Microscopy Unit of the Institute of Biotechnology, University of Helsinki for providing

laboratory facilities. We thank the Core facility for Advanced Light Microscopy, Montebello, Oslo University Hospital, Norway, for access to 3D-SIM. This work was supported by Academy of Finland grants 272130 to Elina Ikonen and Pekka Lappalainen, 282192 to Elina Ikonen, 288475 and 294173 to Markku Varjosalo and 275964 to Simon Pfisterer. Pekka Lappalainen and Elina Ikonen received funding from Sigrid Juselius Foundation. Simon Pfisterer obtained a fellowship from Paulo Foundation. Peter Horvath was supported by the TEKES FiDiPro Programme.

## Author contributions

S.P. designed and performed experiments, analysed data and wrote the manuscript, G.G. performed *in vitro* actin experiments and analysed data. P.H. designed automated lipid droplet analysis tools. S.J.R. provided patient samples and J.P. performed neutrophil isolations. V.T.S. performed 3D-SIM imaging and L.K. electron microscopy. M.V. performed lipid droplet proteomics. P.L. designed experiments and analysed data, E.I. designed experiments, analysed data and wrote the manuscript. All authors commented on and approved the manuscript.

## Additional information

**Competing interests:** The authors declare no competing financial interests.

**Publisher's note**: 

