## [Peer Review File · Nature Communications]

Reviewers' Comments:

Reviewer #1 (Remarks to the Author)

The manuscript by Dr. Ikonen and co-workers describes a novel molecular pathway that controls cellular lipid storage. Authors found actomyosin structure mediated by NMIIa and FMNL1 are localized on LDs and responsible for the dissociation of clustered LDs. Depletion of NMIIa resulted in decreased LD dissociation, a higher cellular TG level and enlarged LD sizes. FMNL1 is responsible for actin filament assembly on LDs. They conclude that cellular actomyosin structure controls lipid storage by regulating LDs dissociation and lipid hydrolysis.

The revisions addressed the concern regarding images quality compared with the previous manuscript. The resolution of images improved a lot, optical sections were used, and statistic analysis of the localization of NMIIa or FMNL1 were performed. Overall, the manuscript is very interesting, the concept is new and the data are of high quality

However, there are still a few concerns regarding the function of NMIIa/FMNL1 and the model proposed by the authors. Specific comments are as follows:

1. It is important to give a statistical analysis in Figure 2d and Figure 4d.
2. In figure 1a/c, besides the enlargement of LDs, a significant decrease in the number of LDs in cells transfected with siNMIIa was observed. In figure 1e, expression of NMIIa in siNMIIa cells increased LD numbers. Therefore, the up-regulation of coalescence/fusion of LDs was a more reasonable explanation for the phenotypes. Disruption of lipolysis would be a secondary effect.
3. Figure 5f/g showed that the number of LDs in MYH9RD patients' cells were significantly increase, whereas the size distribution of LDs were similar in both groups. This is not consistent with authors observation in U2OS cells and can not be explained by the disruption of detachment caused by NMIIa depletion. In addition, two clinical samples in each group may not be able to draw a clear conclusion as the variation is very big. Overall, clinical observation in this figure did not support the conclusion.
4. In Figure 3c/d, why does the number of LD clusters and the number of LDs per cluster decreased in NMIIa KD cell, if NMIIa KD abolished the dissociation of LDs (Figure 3c/d)?
5. The dramatic increase of cellular TG in NMIIa depleted cells was impressive. Therefore, cellular TG levels should also be measured in FMNL1 KD and in Cytochalasin D or Blebbistatin treated cells.
6. If FMNL1 determines the LD localization of NMIIa (Figure 4g/h/i), the authors should give an explanation why does the enhancement of LD sizes in FMNL1 KD cells looks so weak compared with that in NMIIa cells as only small percentage of LDs showed this phenotypes (Figure S5j).

Reviewer #2 (Remarks to the Author)

In the revised version of the manuscript, the authors addressed many of the points raised in my initial review for Nature Cell Biology. The conclusions are clearly more solid and the manuscript has been significantly improved.

However, I have several points that should be addressed or clarified, especially regarding the new data presented in Figure 4 d-f.

1- Figure 2a: the contrast/brightness of NMHIIa in the grey level picture and in the corresponding overlay picture (green channel) is clearly different. The same is true in Figure S1i.

Each original picture of the paper should be checked to be assured that this issue does not occur elsewhere.

2- Figure 4d-j: there are enormous blebs of FMLN1 and LifeAct. These blebs are much bigger than the dots described in Figure 4c. Moreover, I don't see any fission event at time 158sec.

Conversely, one can observe a bleb of FMLN1/LifeAct without LD at time 54sec. Thus, these large fluorescent structures likely result from artifacts of overexpression. This is a central point of the paper and should be clarified with more convincing movies if the authors want to claim the temporal sequence: LD/FLMN-actin/MyosinII/fission.

3- Figure S1k: labels HPOS/MLC are inverted in the overlay picture.

4- Figure S4c is not really meaningful since the LD is already fissioned at time 0. Similarly, there is apparently no fission event in Figure S5h. It would be more convincing to show staining of LD during the fission process like in Figure 2d.

This also raises the following question: could it be that the major role of actin/myosin II is actually to prevent LD refusion rather than driving the fission? This should be at least discussed.

Reviewer #3 (Remarks to the Author)

Reviewer #3 was asked to only comment on the microscopy data

In the revised version of the manuscript, the authors include both new optical and new electron microscopy data of substantially better quality than in their first submission. In addition, flaws in the image processing and data analysis have been corrected. Thus, the authors have addressed all my concerns in a satisfactory manner and I believe that the manuscript can be published in Nature Communications.

Responses to Reviewers' comments:

Reviewer #1:

The manuscript by Dr. Ikonen and co-workers describes a novel molecular pathway that controls cellular lipid storage. Authors found actomyosin structure mediated by NMIIa and FMNL1 are localized on LDs and responsible for the dissociation of clustered LDs. Depletion of NMIIa resulted in decreased LD dissociation, a higher cellular TG level and enlarged LD sizes. FMNL1 is responsible for actin filament assembly on LDs. They conclude that cellular actomyosin structure controls lipid storage by regulating LDs dissociation and lipid hydrolysis.

The revisions addressed the concern regarding images quality compared with the previous manuscript. The resolution of images improved a lot, optical sections were used, and statistic analysis of the localization of NMIIa or FMNL1 were performed. Overall, the manuscript is very interesting, the concept is new and the data are of high quality

We sincerely thank the Reviewer for the thoughtful comments and suggestions that helped us improve this manuscript. We address the remaining concerns below, the corresponding changes in the manuscript are highlighted with yellow.

However, there are still a few concerns regarding the function of NMIIa/FMNL1 and the model proposed by the authors. Specific comments are as follows:

1. It is important to give a statistical analysis in Figure 2d and Figure 4d.

This is a very good point. Figure 2d showed NMIIa and Figure 4d FMNL1 association with LDs. We have now acquired several additional high quality videos of the associations of NMIIa and FMNL1 on clustered LDs, and performed statistical analysis on those (new Fig. 4d-g, new Supplementary Movies 8, 9). We found that about 65% of dissociating LDs harbored FMNL1 and 50% NMIIa, suggesting that these are not rare events. In addition, we found that in most cases, FMNL1 and NMIIa were concentrated on or immediately adjacent to the LDs already before the clustered LDs started to be pulled apart. These findings suggest that the actomyosin structures identified may constitute part of the machinery that helps to dissociate the LDs from one another. However, we wish to point out that our data do not rule out the possibility that the proteins are helping to keep LDs from (re)fusing together (see point 2, and also response to reviewer 2 point 4).

2. In figure 1a/c, besides the enlargement of LDs, a significant decrease in the number of LDs in cells transfected with siNMIIa was observed. In figure 1e, expression of NMIIa in siNMIIa cells increased LD numbers. Therefore, the up-regulation of coalescence/fusion of LDs was a more reasonable explanation for the phenotypes. Disruption of lipolysis would be a secondary effect.

We agree with the Reviewer, and have now included this interpretation in the manuscript. Indeed, it is possible that increased LD coalescence upon NMIIa silencing is a primary event due to loosing an inhibitory effect of the actomyosin structures on LD fusion. With this in mind, we have also altered the title of the manuscript, to refer to lipid droplet dynamics in more general terms.

3. Figure 5f/g showed that the number of LDs in MYH9RD patients' cells were significantly increase, whereas the size distribution of LDs were similar in both groups. This is not consistent with authors observation in U2OS cells and can not be explained by the disruption of detachment caused by NMIIa depletion. In addition, two clinical samples in each group may not be able to draw a clear conclusion as the variation is very big. Overall, clinical observation in this figure did not support the conclusion.

The Reviewer is right in that the LD phenotypes of MYH9RD patients' cells are not identical to those observed in NMIIa silenced U2OS cells and we have now more explicitly stated this in the text (p. 6). We presume that the following (mutually not exclusive) points probably contribute to this. First, the patients' mutation is a one base deletion in the C-terminal non-helical tail of NMIIa that results in a MYH9-RD phenotype (this information has now been included in the text) as well as in the pathognomic NMIIa aggregates in neutrophils (as visualized in the patient cells in Fig. 7a). Therefore, the basis of MYH9RD is more complex than mere reduction of the protein. Second, differences between cell types most probably play a role. For instance, U2OS cells have much more abundant LDs in basal conditions than neutrophils (over 100 vs. about 7). This information has been added to the new Fig. 7c,d. Moreover, since primary neutrophils have a short half-life, we only conducted 1 h oleic acid loading (compared to 7 h loading of NMIIa silenced U2OS cells). Upon 1 h of oleic acid loading, the number of LDs in neutrophils is substantially lower than in U2OS cells, and this may well affect the clustering/dissociation propensities of the LDs.

Despite these differences, we think that it is interesting that primary human patient cells carrying a natural mutation in NMIIa have alterations in their LDs and that these alterations are partly reminiscent of those observed upon NMIIa silencing. Indeed, this is to our knowledge the first report of a lipid imbalance in MYH9-RD patients. To improve this characterization, we have provided additional information on the LDs. Upon similar oleic acid loading, the patient cells generate more LDs than control cells, implying differences in the way oleic acid is deposited in LDs (shown in the new Fig. 7d); moreover, there is in fact a tendency towards increased LD sizes in the patient cells (shown in the inset of Fig. 7e), analogously to the situation in NMIIa depleted cells.

We agree with the Reviewer that two patient samples is a small number. However, in the case of an extremely rare monogenic disease with >70 known different disease mutations it was not feasible to collect a large number of patient samples representing the same mutation.

Nevertheless, we think that the difference between the patient and control neutrophils is conspicuous and merits reporting. We also included a discussion on how lipid imbalance in neutrophils might be pathophysiologically relevant (results, page 6).

4. In Figure 3c/d, why does the number of LD clusters and the number of LDs per cluster decreased in NMIIa KD cell, if NMIIa KD abolished the dissociation of LDs (Figure 3c/d)?

We think that this could be due to a situation where the lack of LD dissociation in NMIIa knockdown cells is balanced by an increased coalescence of the LDs, either via direct fusion and/or lipid transfer. This would result in an increased size heterogeneity of LDs and a net consumption of LDs within a cluster, with a parallel reduction of clusters. We have now discussed these aspects more thoroughly in the manuscript (page 7).

5. The dramatic increase of cellular TG in NMIIa depleted cells was impressive. Therefore, cellular TG levels should also be measured in FMNL1 KD and in Cytochalasin D or Blebbistatin treated cells.

We have now added information on cellular TG levels in FMNL1 and blebbistatin treated cells, as requested. There is a pronounced increase in TGs also in blebbistatin treated cells (new Fig. 1h), as expected. Interestingly, in FMNL1 knockdown cells, the situation is different: TGs are decreased (new Supplementary Fig. 8c). We found that this is due to the fact that FMNL1 silencing also inhibited fatty acid uptake into cells (while NMIIa silencing did not; new Supplementary Fig. 8b). This is not surprising considering that FMNL1 is a multifunctional actin nucleator, also implicated e.g. in phagocytic uptake (Seth, A. et al, Autoinhibition regulates cellular localization and actin assembly activity of the diaphanous-related formins FRLalpha and mDia, J. Cell Biol. 174, 701-13, 2006).

6. If FMNL1 determines the LD localization of NMIIa (Figure 4g/h/i), the authors should give an explanation why does the enhancement of LD sizes in FMNL1 KD cells looks so weak compared with that in NMIIa cells as only small percentage of LDs showed this phenotypes (Figure S5j).

Thank you for pointing this out, it relates to our reply to the previous point. Although FMNL1 regulates the recruitment of NMIIa to LDs, it also affects fatty acid uptake (new Supplementary Fig. 8b). With reduced fatty acid uptake into FMNL1 knockdown cells, the increase in LD size is less pronounced than in NMIIa silenced cells.

Reviewer #2:

In the revised version of the manuscript, the authors addressed many of the points raised in my initial review for Nature Cell Biology. The conclusions are clearly more solid and the manuscript has been significantly improved.

We very much appreciate the Reviewer's efforts to help to improve our manuscript by the constructive suggestions. Below, we have addressed the remaining points. The corresponding changes in the manuscript are highlighted with yellow.

However, I have several points that should be addressed or clarified, especially regarding the new data presented in Figure 4 d-f.

1- Figure 2a: the contrast/brightness of NMHIIa in the grey level picture and in the corresponding overlay picture (green channel) is clearly different. The same is true in Figure S1i.

Each original picture of the paper should be checked to be assured that this issue does not occur elsewhere.

Thank you for the remark. We have now carefully scrutinized all the panels in the grey and overlay images, and adjusted the contrast/brightness so that they are similar. We have also correspondingly readjusted the intensity quantifications of the line profiles in these images. This did not alter the conclusions.

2- Figure 4d-j: there are enormous blebs of FMLN1 and LifeAct. These blebs are much bigger than the dots described in Figure 4c. Moreover, I don't see any fission event at time 158sec. Conversely, one can observe a bleb of FMLN1/LifeAct without LD at time 54sec. Thus, these large fluorescent structures likely result from artifacts of overexpression. This is a central point of the paper and should be clarified with more convincing movies if the authors want to claim the temporal sequence: LD/FLMN-actin/MyosinII/fission.

We agree: the large fluorescent structures are probably related to the high levels of protein expression. We have now performed additional live cell imaging experiments to produce more convincing movies. We present these data in the new Figures 4 and 5, new Supplementary Movies 8, 10-12, and Supplementary Fig. 6. We have also included quantitative analysis of FMLN1 and NMIIa association between dissociating LDs from 130-150 LDs per condition (in the new Fig. 4). We found that in most cases, FMNL1 association with clustered LDs is observed prior to the dissociation of LDs, and the same holds true for NMIIa. In about 30% of dissociating LDs, NMIIa is first observed on LDs once they are being pulled apart. Because of these observations, we favor the interpretation that FMNL1 localizes to sites of LD dissociation to nucleate actin filaments functionalized by NMIIa, and thereby providing a force to assist in LD dissociation. On the

other hand, the observations are also compatible with the idea that FMNL1-actin-myosin II recruited to LDs prevents their (re)association or fusion, as suggested by the Reviewer (point 4).

3- Figure S1k: labels HPOS/MLC are inverted in the overlay picture.

Thank you, this has been corrected.

4- Figure S4c is not really meaningful since the LD is already fissioned at time 0. Similarly, there is apparently no fission event in Figure S5h. It would be more convincing to show staining of LD during the fission process like in Figure 2d.

This also raises the following question: could it be that the major role of actin/myosin II is actually to prevent LD refusion rather than driving the fission? This should be at least discussed.

Thank you for bringing this up. As discussed above, we have now provided additional live cell imaging material that better captures the dissociation events (Fig. 4, 5, Supplementary Movies 8, 9, 10, 11, 12). Please note that we do not use the term LD fission in the manuscript, because we cannot be sure in the case of the smallest LDs when/if they actually undergo fission. We can only reliably monitor their dissociation or getting pulled apart from each other, when we observe the appearance of intervening (non-LD fluorescent) pixel(s) between them. Interestingly, we also observe that dissociated LDs may undergo reassociations and that FMNL1 may be associated with LDs during such events. Therefore we have now also discussed the potential alternative interpretation that a major role of actin/myosin II is actually to prevent LD refusion (Discussion page 7). We have also changed the title of the manuscript, to leave space for this interpretation.

Reviewer #3:

In the revised version of the manuscript, the authors include both new optical and new electron microscopy data of substantially better quality than in their first submission. In addition, flaws in the image processing and data analysis have been corrected. Thus, the authors have addressed all my concerns in a satisfactory manner and I believe that the manuscript can be published in Nature Communications.

Reviewers' Comments:

Reviewer #1 (Remarks to the Author)

Authors addressed most of the concerns. However, the explanation for the clinical data is still not convincing enough to support the conclusion.

Reviewer #2 (Remarks to the Author)

The new movies and interpretation are much more convincing in the revised manuscript. The authors answered all my remaining requests. I therefore recommend publication in Nature Communications.